# Knowledge and Expectations Regarding Sustainable Food Systems among Students from Georgian Agricultural Universities and Georgian Food Industry Representatives

**Allan Estandarte [1,\*], Tekla Gurgenidze [2], Teo Urushadze [2] and Angelika Ploeger [1]**

[1]  Faculty of Organic Agricultural Sciences, University of Kassel, 37213 Witzenhausen, Germany; a.ploeger@t-online.de
[2]  School of Agricultural and Natural Sciences, Agricultural University of Georgia, 0159 Tbilisi, Georgia; teklagurgenidze19@gmail.com (T.G.); t.urushadze@agruni.edu.ge (T.U.)
\*  Correspondence: uk087919@ad.its.uni-kassel.de; Tel.: +33-75-5211804

**Abstract:** This study establishes the role of sustainability in higher education (HE) and the food industry in Georgia by examining Sustainable Food Systems (SFS) background knowledge among students and food industry representatives, their behaviours as consumers, and their level of food citizenship. This study also investigates the most interesting SFS topics in relation to future training, students' expectations in developing competencies, and the SFS elements they deem most important. This cross-sectional study was performed through an online survey comprising a higher education questionnaire administered in five agricultural universities in Georgia which obtained 321 responses. Another questionnaire administered to Georgian food industry (FI) representatives obtained 54 responses. Data were analysed through non-parametric and multivariate statistical analysis. Georgian students and food industry representatives were knowledgeable on Sustainable Food Systems topics, yet some were neither interested nor had received training previous training in SFS. Students' food purchasing and consumption motivations are most influenced by taste and health, demonstrating significant differences between universities. The maintenance of healthy ecosystems was the most important component of SFS, while organic agriculture and agroecology are the most interesting topics. In Georgia, higher education and the food industry both play equally essential roles in the development of Sustainable Food Systems.

**Keywords:** Georgian higher education; teaching and training; European food systems; food systems transformation; sustainability education

## 1. Introduction

The global food system is described as a multifaceted, complex nexus incorporating environmental, economic, social, and technological processes involving food, from its production and utilization to waste disposal [1,2]. To achieve sustainable development at the European level, transitioning towards Sustainable Food Systems is equally imperative for developed and developing countries such as Georgia. In the past, with post-Soviet Georgian governments, the development of the agriculture sector was unprioritized and this was attributed to the absence of a functioning agricultural research–education–extension system in the country [3]. In order to address this gap, since the turn of the millennium, Georgia has been undergoing reforms and transformations through various developments, integrating modernization and the concepts of economic liberalization [4,5].

Traditional education environments focusing on simply increasing environmental knowledge have been found to be inadequate in fostering pro-environmental and sustainable change [6]. Education should not only involve a transfer of knowledge but a space for dialogue to increase the creativity of individuals and groups [5] (p. 1088). In this light, higher education institutions (HEIs) in Georgia play an invaluable role in the formation

of future scholars and professionals, actors who will directly structure and influence the future food system. These actors need to possess a keen attention to system elements and the capacity to make decisions in the face of complexity through various forms of thinking and performance [7]. Georgian HEIs should be able to develop these actors' creativity in solving problems through a systems approach (e.g., transdisciplinary research) instead of narrowly defined and isolated disciplines. This would not only prevent graduates from being insufficiently prepared to deal with food system complexity but also in interacting with multidisciplinary environments [7,8]. Georgian HEIs play a pivotal role in instilling a systems approach through the cognitive, socio-economical, and behavioural domains [7] whilst engaging students to be active food citizens in fulfilling the Sustainable Development Goals (SDGs).

Furthermore, food citizenship, a vital component of SFS development, can be conceptualized as an outcome of transformations in the personal, cultural, political, and practical spheres. Food citizenship is defined as the practice of engaging in food-related behaviours acknowledging the need to move beyond food as a commodity and individuals as consumers [7,9,10]. HEIs play a crucial role in developing students' food citizenship by deepening their grasp of SFS topics through immersive and multi-perspective approaches [7,11,12].

To equally cover the political and practical spheres of transformation, where the "outcomes" and "systems and structures" of transformations are respectively situated [13], mapping out sustainability and its components in the Georgian food industry is equally indispensable. These non-state actors play a significant role in food politics, particularly in the context of the creation and implementation of private norms, rules, and standards which are vital in reforming the food system [14].

Georgia's geographical setting allows it to be a country of rich agrobiodiversity and this provides great potential for economic development [5,15]. In the past two decades, the Biological Farming Association Elkana, in partnership with German public institutions, spearheads strategies in the development of organic farming by providing advisory services to farmers, leading to the country's harmonization with EU regulations in the present day [16]. However, Georgia still faces challenges in preserving its natural biodiversity while developing policies to accelerate economic development and population well-being, as reported by previous studies [5,16]. Furthermore, there is very limited literature examining the role of Georgian higher education in the empowerment of future generations to address these challenges and thereby ensure sustainable development.

Given this context, the following working hypotheses serve as the basis for this research paper:

**Hypothesis 1 (H1).** *Georgian students and food industry representatives may have limited background knowledge of SFS and its topics. The authors propose this hypothesis in the assumption that the level of knowledge of the Georgian population may be lesser in comparison to the western European population, where a sizeable proportion of students enrolled in food and agriculture programs reported not receiving courses on Sustainable Food Systems or courses covering related topics [12]. Another study carried out in a neighbouring Asian country, Pakistan, reported that both students and educators have inadequate knowledge of SFS [17]. Furthermore, the limited number of scientific publications from Eastern European research bodies and universities suggests that Sustainable Food Systems may be less developed in these regions [18].*

**Hypothesis 2 (H2).** *Students' behaviours as consumers strongly influence their level of food citizenship (motives, values, and habits toward food). Emerging studies establish that consumers and citizens play an essential role in sustainability transition [9,12,19]. Concurrently, factors that influence consumer behaviour, such as social background, family cooking traditions, and cultural and financial background, have been established as influencing students' levels of food citizenship [12].*

**Hypothesis 3 (H3).** *The element of SFS that is most important for students and food industry representatives is "maintains healthy ecosystems".*

**Hypothesis 4 (H4).** *The teaching methods preferred and most interesting to students which should be included in contemporary education programs are "seminars and interactive workshops" while "e-learning courses" are considered least interesting.*

The authors postulate H3 and H4 based on the results of a similar study performed in western European universities where "healthy ecosystems" was rated as the most important element of SFS, while "seminars and interactive workshops" and "e-learning courses" are considered the least interesting [12]. Additionally, Georgia has received support from experts and lecturers from German public institutions, such as the University of Kassel and the German Federal Agency for Nature Protection, in training stakeholders on sustainable farming practices (i.e., organic composting) [20]. This suggests that topics of interest and teaching methods in SFS may be shared in both populations.

**Hypothesis 5 (H5).** *The topics related to SFS and the Georgian agri-food industry that are most relevant and interesting for future training are "organic agriculture" and "agroecology". Spearheaded by Elkana, the Georgian transition to SFS involves strategies that mainly focus on organic agriculture and the protection of the environment over the previous two decades [20]. Additionally, unsustainable agricultural practices which led to the deterioration and depletion of natural resources remain a present challenge [4]. Therefore, these topics are hypothesized as the most interesting for Georgian stakeholders.*

## 2. Materials and Methods

### 2.1. Study Design and Methods

A two-phase cross-sectional quantitative study design was implemented to come up with a more holistic picture of SFS in Georgia. The first phase involved a structured survey adapted from the European Union (EU) SUSPLUS project (Innovative Education towards SFS) in which the working group was part of and was modified to serve the Georgian higher education (HE) system. This Higher Education Questionnaire (HEQ) included 24 questions (composed of open-ended questions, multiple-choice questions, and Likert scales) and was organized into three major components. The first part was "present attitudes, values, and behaviours", which inquired about students' purchasing habits (frequency of food purchasing and food preparation), lifestyle, and the factors influencing their food citizenship. The second part involved their "present knowledge and understanding" of the topic of SFS and its elements, which included asking students about their overall learning experience with SFS and which specific SFS topics they had covered in their modules. The third part inquired about students' expectations for future higher educational curricula (which skills were most valuable for them, which topics they found most interesting, and what kind of teaching methods they preferred). The HEQ was administered to all universities in Georgia offering agricultural/horticultural science and food/nutrition science programs at various levels (bachelor's, master's, and Ph.D.) as illustrated in Table 1. Adapted from the methods of Migliorini et al. (2020), the protocol prioritized students in agri-food universities since the sensitisation of this young generation of future active consumers and agri-food professionals to the topic of Sustainable Food Systems is of vital importance [7]. All students who pursued other degree programs and those who were not enrolled in the listed universities were excluded from the analysis.

**Table 1.** Georgian agricultural universities and their topics of focus.

| University Name | Acronym | Focus Areas |
| --- | --- | --- |
| Agricultural University of Georgia | "AUG" | "Agriculture, agronomy, agricultural engineering, ecology, food science, viticulture, veterinary sciences" |
| Akaki Tsereteli State University | "ATSU" | "Medicine, agricultural sciences, natural sciences" |
| Batumi Shota Rustaveli State University | "BSRSU" | "Agriculture, agricultural engineering, agro-technology, food science, and forestry" |
| Georgian Technical University | "GTU" | "Business, law, engineering, architecture, agricultural sciences, and biosystems engineering" |
| Telavi Iakob Gogebashvili State University | "TIGSTU" | "Agrarian sciences, educational sciences, humanities, social sciences, and law" |

On the other hand, the second phase was a survey composed of 19 questions administered to food industry representatives in Georgia. This included industry partners, farmers, professionals in associations, and representatives of academia. Similar to the HEQ, the food industry questionnaire (FIQ) was also composed of three components. The first part inquired about "present attitudes, values, and behaviours" on SFS topics and how these individuals find these factors relevant to their line of business or profession. The second part inquired about their "background knowledge and understanding of SFS and related topics" (their professional training in SFS). The third part asked industry representatives about their "future expectations about SFS in an industrial context" (professional development, desirable skills for future professionals, topics for future training). Both questionnaires were translated into Georgian and were administered online using Survey Sparrow with a data collection period from June to August 2021. The survey was available both in English and Georgian, and respondents were able to choose the language while accomplishing the questionnaires. To gather a sufficient spread of students across the five universities, the survey link was shared through social media, emails, and word of mouth, with the assistance of the Agricultural University of Georgia (Prof. Dr. Teo Urushadze) in following up on responses and cascading the online link for the survey until the representative sample size was achieved.

The first phase of the study had a total of 321 responses collected from five Georgian universities, namely, the Agricultural University of Georgia (AUG) N = 117, Akaki Tsereteli State University (ATSU) N = 86, Batumi Shota Rustaveli State University (BSRSU) N = 31, Georgian Technical University (GTU) N = 57, and Telavi Iakob Gogebashvili State University (TIGSTU) N = 29. All the participants were presently enrolled in their respective Georgian universities.

The second phase of the study was initiated after gathering sufficient responses for the first phase (students). Through the assistance of the Agricultural University of Georgia and its network of industry partners and academics, a total of 54 responses were again collected using b snowballing and sending the survey link online via mail and social media platforms. The respondent pool consisted of professionals from various food industries and sectors of the value chain, including academics, growers and producers, retailers, and food industry professionals.

*2.2. Statistical Analysis*

Using IBM SPSS Statistics 26, both data sets for phase 1 and phase 2 were analysed to be not normally distributed, as commonly observed among questionnaires using the Likert scale. The Kruskal–Wallis H non-parametric test was performed to assess significant differences for each categorical variable. Statistically significant differences ($p \leq 0.05$) were observed between the Georgian universities. Due to the limited size for phase 2, differences between categorical variables were observed to be not significant. To analyse the relationship between responses, one-way ANOVA with Games–Howell post hoc tests

was performed. Furthermore, principal component analysis and Pearson correlations were performed to study the relationship between students' responses from different universities.

## 3. Results and Discussion

### 3.1. Demographic Data of Respondents: Higher Education and Food Industry Surveys

3.1.1. Demographic Data of Georgian University Students

A representative sample size of 321 respondents distributed at the 5 universities was obtained and illustrated in Table 2. The sample was 46% male, 53% female, and 1% identified as non-binary. The highest proportion of females was observed in the Agricultural University of Georgia (AUG) at 65%, while the lowest was observed in Batumi Shota Rustaveli State University (BSRSU) at 40%. The average respondent age was 22 years old, ranging from 18 to 52 years. Most of the students studied agricultural and horticultural sciences (70.7%), while 29.3% studied food and nutrition sciences. Studies in bachelor's, master's, and Ph.D. degrees were 80%, 15%, and 5%, respectively. Among the participants, 18% were first-year students, 32% were second-year students, 23% were in their third year, and 27% were in their fourth year or more.

**Table 2.** Demographic data of Georgian students.

|  | Bachelor's | Master's | Ph.D. or Higher | Total |
|---|---|---|---|---|
| Number of students | 256 | 49 | 16 | 321 |
| Gender ratio (male: female) | 1.032 | 0.361 | 0.6 | 0.865 |
| Mean age in years | 20.64 | 24.51 | 33.25 | 21.86 |

3.1.2. Demographic Data of Georgian Food Industry Survey

Of the 54 respondents who participated in the Food Industry Survey, 63% identified as female while 37% identified as male. As shown in Table 3, in terms of educational background, the highest proportion (43%) of respondents attained a master's degree, followed by those with a bachelor's degree (31%), and those having a Ph.D. or a higher qualification (26%). Most of the respondents were either employed in primary production (27.8%), academia or research (20.4%), or retail or distribution (16.7%). In terms of the food business category, most respondents were engaged in fruits and vegetables (29.6%), alcoholic beverages (16.7%), and dairy or cheese (11.1%).

**Table 3.** Demographic data of Georgian food industry representatives.

|  | Bachelor's | Master's | Ph.D. or Higher | Total |
|---|---|---|---|---|
| Number of respondents | 17 | 23 | 14 | 54 |
| Gender ratio (male: female) | 0.417 | 0.769 | 0.56 | 0.865 |
| Mean age in years | 33.29 | 39 | 49.79 | 40 |

3.1.3. Students' Background Knowledge of Sustainable Food Systems and Its Topics

Results for the student's background knowledge of Sustainable Food Systems showed that 58% of the students were interested in the topic of SFS, while 46% reported (AUG 60%, TIGSU 48%, GTU 39%) that they had never taken a course in their program which covered SFS and its topics. As illustrated in Table 4, ATSU and BSRSU students reported that most of the topics were covered in their programs in comparison to AUG, GTU, and TIGSU students. In contrast to a highly specialized university in the agricultural sciences such as AUG, ATSU, which is located in the west region of Georgia, ranked the highest in most of the topics, such as "Traditional/regional food", "Community-supported agriculture", "Food box schemes", "Food sovereignty", "Vegetarianism", "Veganism", and "Food loss and waste", with significant differences among other universities. AUG and BSRSU cover

the topic "Food safety" more fully than other universities. BSRSU covers "Food security" and "Sustainable Development Goals" the most in comparison to other universities.

**Table 4.** Sustainable Food Systems topics are covered in different study programmes among five agricultural universities in Georgia.

| | | | | AUG | ATSU | BSRSU | GTU | TIGSU | |
|---|---|---|---|---|---|---|---|---|---|
| Variable | Total Mean | N | SD | 117 | 87 | 31 | 57 | 29 | *p*-Value |
| Traditional food/regional food (PDO or PGI) | 1.62 | 321 | 0.69 | 1.53a | 1.94abc | 1.68 | 1.39b | 1.41c | *** |
| Community- supported agriculture (CSA) | 1.61 | 321 | 0.69 | 1.46a | 1.94abc | 1.68 | 1.46b | 1.45c | *** |
| Food box schemes | 1.45 | 321 | 0.64 | 1.43 | 1.66ab | 1.45 | 1.3a | 1.17b | ** |
| Food sovereignty | 1.54 | 321 | 0.66 | 1.47 | 1.72 | 1.65 | 1.40 | 1.41 | |
| Food security | 2.12 | 321 | 0.74 | 2.08 | 2.24 | 2.29 | 2.07 | 1.90 | |
| Food safety | 2.25 | 321 | 0.74 | 2.31 | 2.18 | 2.42 | 2.19 | 2.17 | |
| Sustainable Development Goals (SDGs) | 1.72 | 321 | 0.71 | 1.58ab | 1.92a | 2b | 1.65 | 1.55 | ** |
| Vegetarianism | 1.32 | 321 | 0.56 | 1.23a | 1.49ab | 1.42 | 1.26 | 1.21 | ** |
| Veganism | 1.30 | 321 | 0.56 | 1.2a | 1.52ab | 1.29 | 1.25 | 1.17b | *** |
| Food loss and waste | 1.84 | 321 | 0.69 | 1.67a | 1.99a | 1.97 | 1.89 | 1.79 | |

Notes: Respondents chose between 1 = Not at all covered, 2 = Yes, there were a few lectures (1–4) on this topic within other courses, 3 = Yes, it was a whole course (at least 15 h). Kruskal–Wallis tests: ** marks *p* < 0.001; *** marks *p* < 0.0001; letters indicate significant differences between universities (post hoc Games–Howell tests). The letters a, b, c signify statistical differences between universities.

*3.2. Food Industry Representatives' Background Knowledge of Sustainable Food Systems and Its Topics*

The largest proportion—46.3% of respondents—indicated that they were interested, while 42.6% reported that they were a little bit interested in Sustainable Food Systems. When asked if they had attended training covering topics on Sustainable Food Systems, 50% reported that they had already received training, while 50% had not received training at all. The lack of interest among the food industry representatives may be explained by the lack of knowledge and experience of topics related to SFS. Furthermore, those who had already attended training on SFS reported that this training was not provided by their employer or the company they were working for. Even with a very limited population for the survey, these findings reveal that, presently, in the Georgian food industry, there is very little awareness of SFS and therefore emphasize the role of higher education institutions in forming future professionals and catalysing this transformation.

*3.3. Students' Behaviour as Consumers and Their Levels of Food Citizenship*

Referring to H2 (How does students' behaviour as consumers affect their food citizenship?), the results for "values and motives for food purchasing and eating" and "food purchasing and cooking frequency" are identified as determinants of students' food citizenship. These findings are illustrated on Table 5.

Values and Motives of Students for Purchasing and Eating

The results presented in Table 5 demonstrated significant differences amongst students' values influencing food purchasing and consumption decisions based on which university they came from. Among the various motives and values, "health" and "taste" were considered most important by the students, while "special diet" and "tropical production" were the least important. ATSU students stand out from the four other universities since its students considered "labels", "seeking tastes from childhood", "environmental", and "social" impacts most important, demonstrating significant differences from AUG, GTU, and TIGSU. Furthermore, students from ATSU considered "tropical production" significantly more important than the four other universities. That ATSU students had the highest

means for several of the values and motives may be attributed to the university's location. The campus being in a more rural and western setting in comparison to Tbilisi, students may have better access to local produce and healthier food choices at more affordable prices. On the other hand, GTU students considered "social impact", "animal welfare", "local and tropical production", "labels", and "seeking tastes from childhood" as not important, scoring lower than the other four universities, while considering "price" of moderately high importance. AUG students also considered environmental impact least among the five universities. This finding is contrary to AUG's strong focus on organic agriculture and its numerous collaborations with Western European universities. Moreover, the high proportion of international students mostly concentrated in Tbilisi may have affected the observed scores. One notably common motivation for all students which was considered of moderately high importance was "price".

**Table 5.** Students' motives and values on food purchasing and eating.

| Variable | Total Mean | N | SD | AUG 117 | ATSU 87 | BSRSU 31 | GTU 57 | TIGSU 29 | *p*-Value |
|---|---|---|---|---|---|---|---|---|---|
| Environmental impact | 2.57 | 321 | 0.54 | 2.42a | 2.74a | 2.58 | 2.60 | 2.66 | ** |
| Health | 2.85 | 321 | 0.36 | 2.75a | 2.93ab | 2.90 | 2.86 | 2.97ab | ** |
| Price | 2.42 | 321 | 0.54 | 2.47a | 2.5a | 2.19 | 2.37 | 2.31 | |
| Social impact | 2.08 | 321 | 0.69 | 2.03a | 2.41abc | 2.10 | 1.81b | 1.86c | *** |
| Taste | 2.68 | 321 | 0.53 | 2.83ab | 2.84cd | 2.61 | 2.28ac | 2.48bd | *** |
| Animal welfare | 2.44 | 321 | 0.60 | 2.31a | 2.74ab | 2.45 | 2.18bc | 2.55c | *** |
| Labels | 2.19 | 321 | 0.75 | 2.15a | 2.58abc | 2.16 | 1.77ab | 1.97c | *** |
| Special diet | 1.77 | 321 | 0.79 | 1.66a | 2.09ab | 1.84 | 1.51b | 1.69 | *** |
| Local production | 2.28 | 321 | 0.71 | 2.3a | 2.56abc | 2.32 | 1.91ab | 2.0c | *** |
| Tropical production | 1.93 | 321 | 0.69 | 1.88ab | 2.31abcde | 1.84c | 1.6d | 1.76e | *** |
| Seeking tastes from childhood | 2.03 | 321 | 0.73 | 1.96a | 2.36abc | 1.84b | 1.82c | 1.93 | *** |

Notes: Respondents chose between 1 = not important, 2 = moderately important, 3 = very important. Kruskal–Wallis tests: ** marks $p < 0.001$; *** marks $p < 0.0001$; letters indicate significant differences between universities (post hoc Games–Howell tests). The letters a, b, c, d, e signify statistical differences between universities.

Examining the relationships among the responses (Figure 1) provided by the students, those who considered labels in their purchasing decisions also considered special diets and animal welfare and strongly considered environmental impact and local and tropical production (in Georgia). These correlations suggest that students who pay attention to labels may be vegans/vegetarians or follow religious practices (fasting and modification of diets) during special periods in the year. Those who cared about animal welfare considered environmental and social impacts on their purchasing decisions. Students who considered "seeking tastes from childhood" in their purchasing decisions considered tropical production. Price, on the other hand, had a weak positive correlation with environmental impact, social impact, and animal welfare while having the strongest correlation with taste among the motives given. This suggests that even though students were wary of price, they still considered the mentioned factors in their purchasing decisions. Tropical production was positively correlated with taste, animal welfare, and special diet. Interestingly, no correlation was observed between health and taste nor between health and price. No negative correlations were observed among the values and motives. There were no significant differences observed between students based on sex or field of study.

Referring to the results of the principal components analysis illustrated in Figure 2, students who were motivated by taste tended to care less about health. Moreover, the food purchasing decisions of students were not strongly influenced by price. In comparison to the results of a similar study performed at 10 universities in the European Union [7], Georgian students also considered health as an important aspect while being less "price-sensitive". This may suggest that Georgian students who can access tertiary education in universities in highly urbanized regions come from families of middle to upper-middle class economic

backgrounds, while underprivileged students, especially those from rural areas, have limited access to these higher educational institutions [21]. Furthermore, the same reference elucidates that state spending on higher education is greater than primary and secondary education, further widening social and economic gaps since fewer disadvantaged students can pursue higher education.

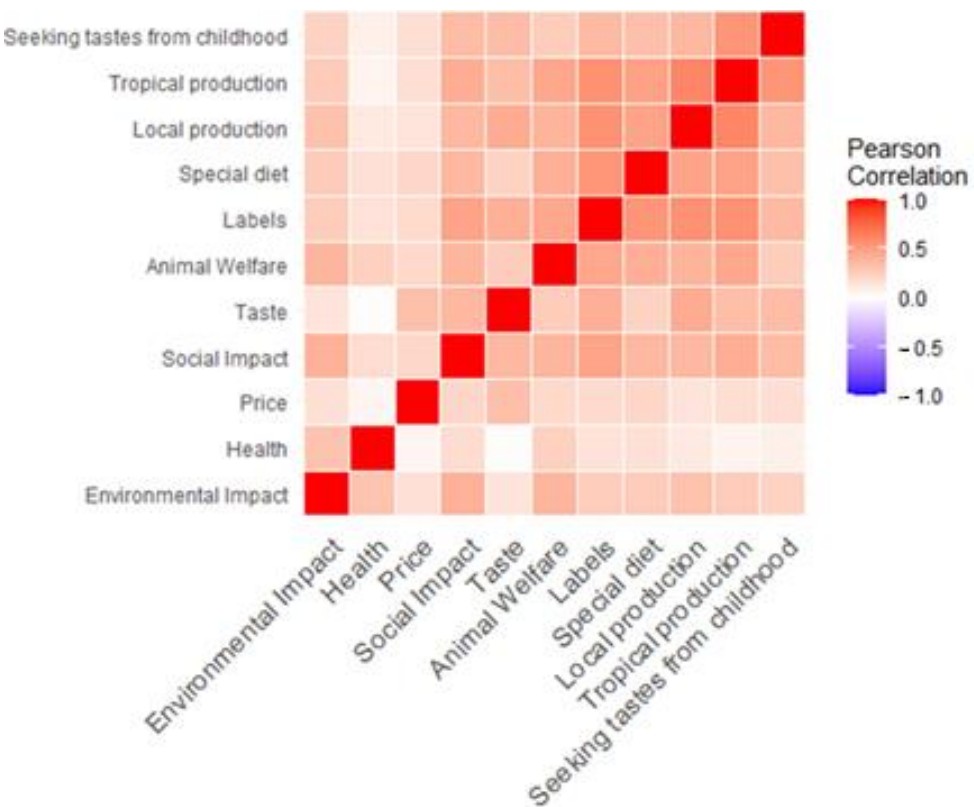

**Figure 1.** Correlation between different values and motives of students for food shopping and eating.

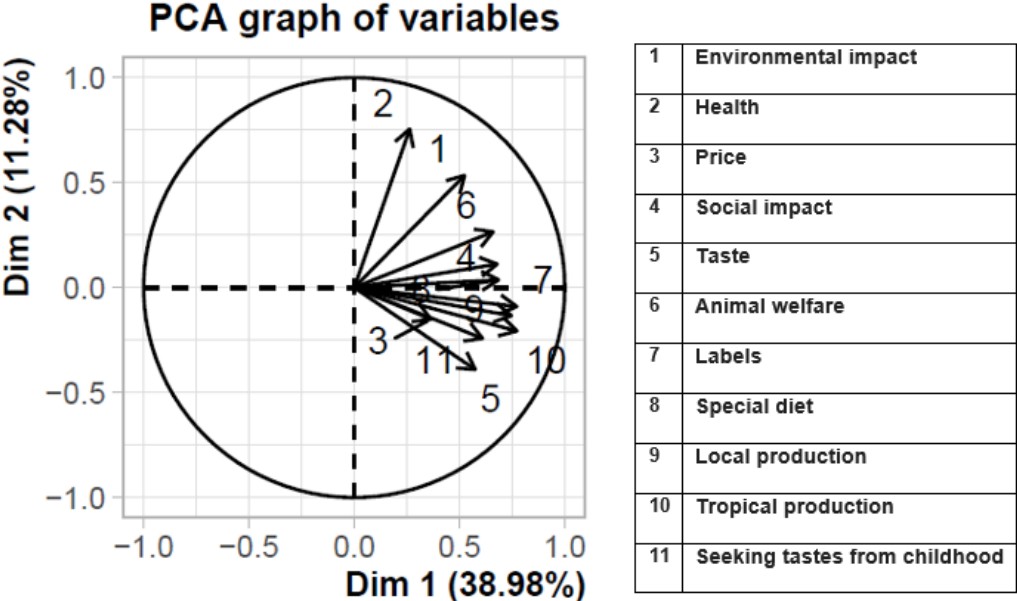

**Figure 2.** Results of principal component analysis of the values and motives of students for food purchasing and eating. Note: The length and direction of arrows denote the magnitude of single questions concerning the first two principal components.

Georgian students are also strongly influenced by labels (Figure 2), confirming the findings of Todua that, generally, Georgian consumers have positive attitudes towards food labelling and are particular about the clarity of the information presented to them such that their purchasing decisions are thereby influenced [22]. In contrast with Western European students [7], Georgian students consider health and taste strongly in their purchasing decisions, yet no correlation was observed between these two values. This implies that the importance of health is a weak predictor of the importance of taste among students. This observation can be explained by the transition to unhealthy diets among students due to the limited availability of food options [21] and the increasing presence of fast-food establishments in Georgian urban areas where schools or universities are situated [22]. Finally, Georgian students considering "seeking tastes from childhood" as an important factor in their purchasing decisions may be explained by the socio-political developments in post-Soviet Georgia during the 1990s. During this period, Muehlfried described eating and participating in banquets as popular, this being encouraged by the numbers of restaurants opening in urban areas, especially in Tbilisi, even under the difficult economic situation [23]. These events were the occasion for families to socialize, go out of their households, and take pleasure from abundant amounts of food. For university students who are away from their families in the course of their studies, these experiences during their childhood may therefore elicit a sense of longing as they miss the tastes from such events.

### 3.4. Students' Food Purchasing and Cooking

Almost a quarter (23%) of the Georgian students who participated in the survey reported that they were in charge of purchasing food for the households they were living in. On the other hand, the majority of the students (48%) participated in food purchasing activities for their household but shared the responsibility with someone else. The frequencies of cooking and food purchasing are illustrated in Figures 3 and 4. A majority of the students participated in both purchasing and cooking food twice or three times a week. As shown in Figures 5 and 6, ATSU and AUG students purchase and cook food the most frequently (between once a week and twice or thrice a week). These scores were observed to be significantly higher in comparison to TIGSU students, who, on average, purchase and cook food between twice or thrice a month and once a month. In general, the majority of the students purchase and cook food either every day or twice/thrice a week. A similar pattern can be observed in the Western European students, especially at the Technical University of Madrid [7], since a majority of Georgian students live with their families. With these established findings, it can be inferred that Georgian students' food purchasing and cooking habits are influenced by socio-cultural backgrounds, financial situation, and traditions, which therefore play an indispensable role in developing their food citizenship. Undoubtedly, universities play a crucial role in developing students' levels of food citizenship by deepening their grasp of SFS topics through immersive and multi-perspective approaches which directly involve them in such complex and dynamic topics [7,11,12].

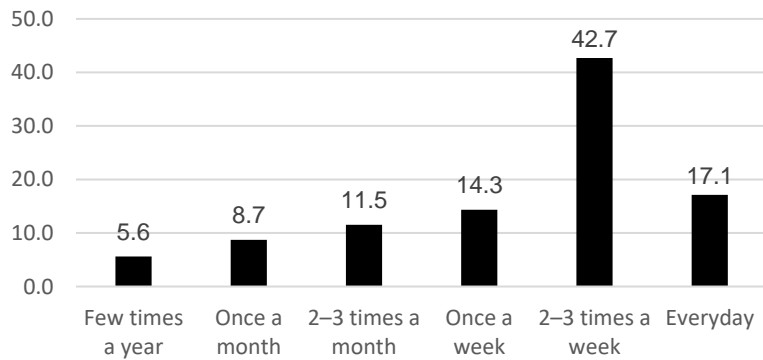

**Figure 3.** Number of students purchasing food for their household.

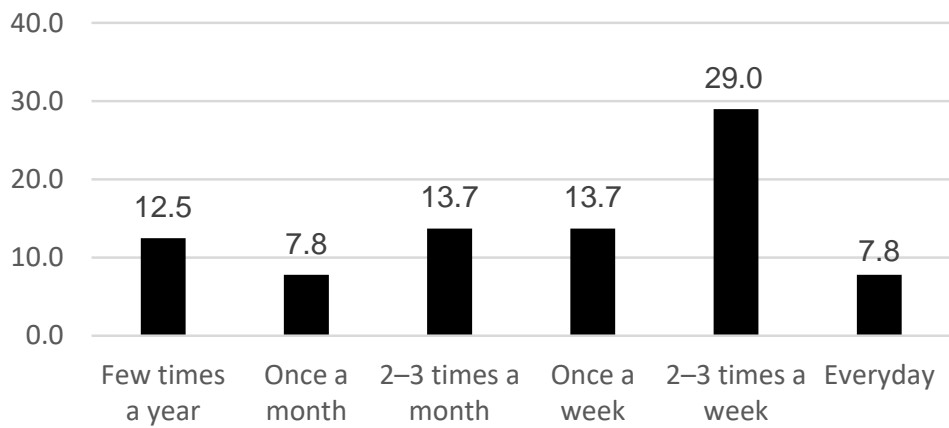

**Figure 4.** Number of students cooking food for their household.

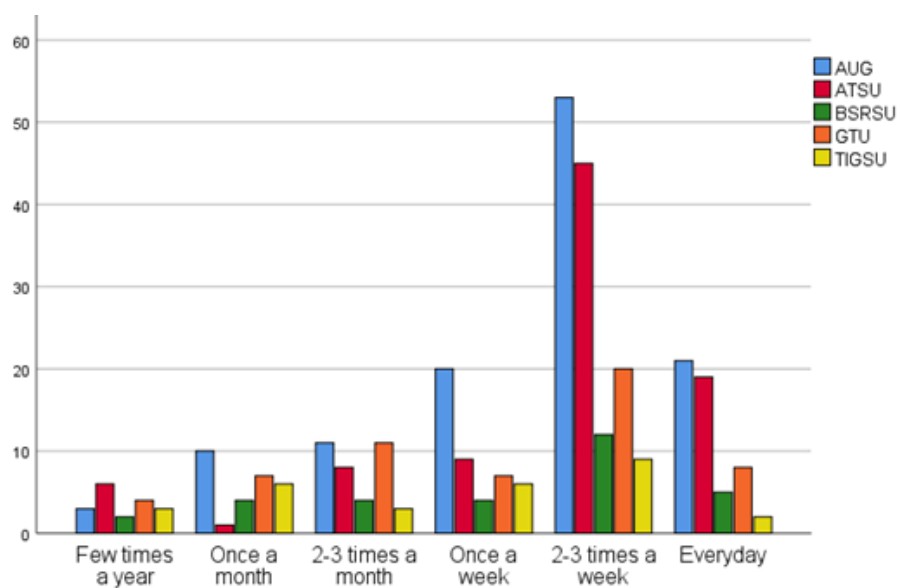

**Figure 5.** Number of times Georgian students purchase food in the household they live per university.

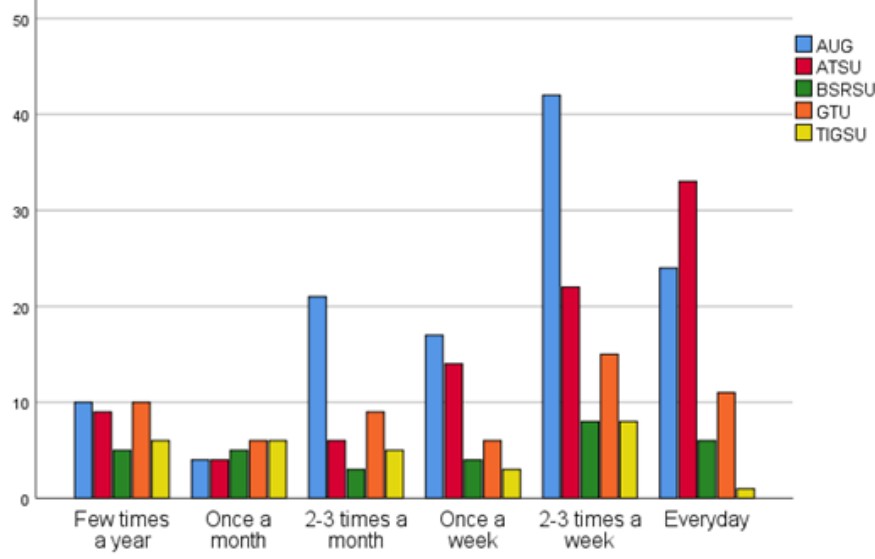

**Figure 6.** Number of times Georgian students cook in the household in which they live per university.

### 3.5. Students' Preferences on Sustainable Food Systems Topics and Elements

All Georgian students rated all the listed elements as important, with "maintains healthy ecosystems" and "protects biodiversity" ranking the two most important, as illustrated in Table 6. Significant differences were observed between the five universities with respect to all elements except "protects biodiversity". AUG and ATSU ranked the SFS elements highest compared to the other universities, exhibiting significant differences from GTU and TIGSU. This may be justified by the fact that SFS topics are not comprehensively covered or that no courses were offered in GTU (39%) and TIGSU (48%), revealing the students' lack of background knowledge of the mentioned topics. Moreover, GTU and TIGSU students were reported to purchase and cook food the least among the five universities, showing significant differences from AUG and ATSU students. This again underlines the need for higher education institutions to provide a strong foundation and deeper understanding of SFS concepts, thus enabling students to acquire more sustainable behaviours as consumers and strengthen their level of food citizenship.

**Table 6.** Students' opinions on the different elements of a sustainable food system.

| Variable | Total Mean | N | SD | Bachelor's 17 | Master's 14 | Ph.D. or Higher 54 | *p*-Value |
|---|---|---|---|---|---|---|---|
| Makes nutritious food available, accessible, and affordable to all | 2.46 | 54 | 0.539 | 2.53 | 2.3 | 2.64 | |
| Maintains healthy ecosystems | 2.56 | 54 | 0.572 | 2.59 | 2.39 | 2.79 | |
| Respects the needs of future generations | 2.54 | 54 | 0.573 | 2.65 | 2.35 | 2.71 | |
| Has minimal negative impact on the environment | 2.35 | 54 | 0.677 | 2.47 | 2.04a | 2.71a | ** |
| Encourages local production and distribution infrastructures | 2.2 | 54 | 0.683 | 2.29 | 1.91a | 2.57a | * |
| Is humane and just, protecting farmers and other workers, consumers, and communities | 2.13 | 54 | 0.754 | 2.18 | 1.87 | 2.5 | * |
| Respects animal welfare | 2.22 | 54 | 0.718 | 2.18 | 2.04 | 2.57 | |
| Is economically sound (provides fair income to producers, distributors, and sellers) | 2.02 | 54 | 0.765 | 2 | 1.74a | 2.5a | * |
| Protects biodiversity | 2.41 | 54 | 0.687 | 2.47 | 2.22 | 2.64 | |

Notes: Respondents chose between 1 = not important, 2 = moderately important, 3 = very important. Kruskal–Wallis tests: * marks $p < 0.01$; ** marks $p < 0.001$; letters indicate significant differences between universities (post hoc Games–Howell tests).

### 3.6. Georgian Food Industry Representatives' Preferences on Sustainable Food Systems Topics and Elements

All the respondents indicated that all elements of SFS are important. The most important SFS elements were "maintaining healthy ecosystems" and "respecting the needs of future generations", while the least important was "is economically sound (provides fair income to producers, distributors, and sellers)". Due to the small sample size of the second phase, only a few significant differences were observed between the respondents' educational attainments. No significant differences were observed between food business categories and food value chain sectors. As illustrated in Table 7, Ph.D. holders rated all of the elements higher, in terms of importance, in comparison to bachelor's and master's degree holders. This may be due to those professionals possessing PhDs being not only more engaged in socio-environmental issues but also more immersed in interdisciplinary sustainability perspectives [24]. Moreover, these elements are considered less important by bachelor's and master's degree holders, which may be attributed to their lack of knowledge or previous training in SFS. These findings reveal that, in Georgia, education on Sustainable Food Systems is still confined to highly specialized instruction and is far from mainstream

pedagogy. This finding again provides a solid basis for the importance of incorporating SFS modules in universities, starting with bachelor's degrees.

**Table 7.** Food industry representatives' opinions on the different elements of a sustainable food system.

| | | | | AUG | ATSU | BSRSU | GTU | TIGSU | |
|---|---|---|---|---|---|---|---|---|---|
| Variable | Total Mean | N | SD | 117 | 87 | 31 | 57 | 29 | *p*-Value |
| Makes nutritious food available, accessible, and affordable to all | 2.54 | 321 | 0.620 | 2.71ab | 2.7cd | 2.42 | 2.25ac | 2.03bd | *** |
| Maintains healthy ecosystems | 2.69 | 321 | 0.510 | 2.8ab | 2.83cd | 2.55 | 2.51ac | 2.34bd | *** |
| Respects the needs of future generations | 2.60 | 321 | 0.610 | 2.72ab | 2.78cd | 2.52 | 2.28ac | 2.28bd | *** |
| Has minimal negative impact on the environment | 2.55 | 321 | 0.630 | 2.73ab | 2.63c | 2.52 | 2.28ac | 2.17b | *** |
| Encourages local production and distribution infrastructures | 2.52 | 321 | 0.670 | 2.73abc | 2.68de | 2.26a | 2.18bd | 2.14ce | *** |
| Is humane and just, protecting farmers and other workers, consumers, and communities | 2.50 | 321 | 0.710 | 2.74ab | 2.68cd | 2.32 | 2.05ac | 2bd | *** |
| Respects animal welfare | 2.49 | 321.00 | 0.610 | 2.53 | 2.68a | 2.29 | 2.3a | 2.31 | *** |
| Protects biodiversity | 2.65 | 321.00 | 0.510 | 2.67 | 2.74 | 2.61 | 2.58 | 2.52 | |

Notes: Respondents chose between 1 = not important, 2 = moderately important, 3 = very important. Kruskal–Wallis tests: *** marks *p* < 0.0001; letters indicate significant differences between universities (post hoc Games–Howell tests).

### 3.7. Students' Preferred Future Learning Topics on Sustainable Food Systems

As reported in Table 8, all the topics mentioned were interesting to the students, with organic agriculture and agroecology ranking the highest. This may be explained by the consistent growth of the organic agriculture industry which has outpaced the whole Georgian food industry in terms of output in recent years [3]. Additionally, the Georgian dairy industry has been relentless in its efforts to meet EU food regulations [5], which justifies the strong interest among university students. Significant differences were observed among the universities except for the topics of organic agriculture, organic food, and agroecology. AUG and ATSU students were most interested in the mentioned topics, while GTU and TIGSU students ranked the least interested on average. This observation may, again, be justified by the lack of knowledge and first-hand experiences of SFS topics amongst GTU and TIGSU students.

**Table 8.** Topics of interest for future teaching courses, as reported by Georgian students.

| | | | | AUG | ATSU | BSRSU | GTU | TIGSU | |
|---|---|---|---|---|---|---|---|---|---|
| Variable | Total Mean | N | SD | 117 | 87 | 31 | 57 | 29 | *p*-Value |
| Organic food | 2.56 | 321 | 0.690 | 2.66 | 2.50 | 2.55 | 2.49 | 2.45 | |
| Fair trade | 2.34 | 321 | 0.690 | 2.48abc | 2.7adef | 2.23d | 1.79be | 1.9cf | *** |
| Slow food | 2.15 | 321 | 0.640 | 2.27a | 2.24 | 1.97 | 2a | 1.93 | * |
| Agroecology | 2.60 | 321 | 0.660 | 2.64 | 2.63 | 2.68 | 2.60 | 2.28 | |
| Organic agriculture | 2.68 | 321 | 0.740 | 2.69 | 2.60 | 2.74 | 2.75 | 2.66 | |
| Protected Denomination of Origin (PDO) and Protected Geographical Indication (PGI) | 2.25 | 321 | 0.740 | 2.42a | 2.48bc | 2.16 | 1.75ab | 1.93c | *** |
| Local food | 2.51 | 321 | 0.710 | 2.68ab | 2.69cd | 2.48 | 2.09ac | 2.1bd | *** |
| Community-supported agriculture (CSA) | 2.28 | 321 | 0.560 | 2.42a | 2.57bc | 2.23 | 1.75ab | 1.93c | *** |
| Food box schemes | 2.14 | 321 | 0.560 | 2.37ab | 2.27c | 2.10 | 1.61ac | 1.86b | *** |
| Sustainable Development Goals (SDGs) | 2.42 | 321 | 0.690 | 2.53a | 2.61b | 2.29 | 2.14ab | 2.14 | *** |

Notes: Respondents chose between 1 = not interesting, 2 = moderately interesting, 3 = very interesting. Kruskal–Wallis tests: * marks *p* < 0.01; *** marks *p* < 0.0001; letters indicate significant differences between universities (post hoc Games–Howell tests).

### 3.8. Food Industry Representatives Preferred Topics for Future Training

The topics most interesting for future training amongst food industry representatives were organic agriculture and agroecology (data not shown). The least interesting topics

were food box schemes and sustainable diets. This finding, again, reveals that the lack of interest in sustainable diets may be attributed to a lack of knowledge or previous training concerning the subject. Sustainable diets are a result and driver of food systems and therefore provide a perspective on the transition towards sustainability [25]. Sensibilization to sustainable diets is crucial for the Georgian food system transition and therefore should be comprehensively covered in food–agricultural programs and as part of continued professional development. In conjunction with the students' preferred topics, organic agriculture and agroecology are considered the primary focus of both the industry and higher education institutions. Confirming the assumptions of Al Sidawi et al. (2020), these common interests and foci between higher education institutions and the private sector provide a promising opportunity in supporting the Georgian value chain by strengthening their relationships. This commonality presents future career opportunities for students and improved support both from the private and public sectors for research and instruction, thereby leading to economic development.

*3.9. Students' Expectations for Future Teaching Programmes for Skills, Topics, and Methods*

A majority of the students (85%) believed that Sustainable Food Systems topics would be useful for their future careers. All learning skills were indicated as highly interesting by students, while significant differences were exhibited between universities (Table 9). The skills ranked highly interesting were creative problem-solving skills, the ability to adapt/act in new situations, the ability to innovate and create, and the ability to make judgements and justify decisions. On the other hand, even with a high average, the least interesting skill for the students was the ability to search for relevant information on the internet.

**Table 9.** Different learning skills of interest to Georgian students.

| SKILLS | Total Mean | N | SD | AUG 117 | ATSU 87 | BSRSU 31 | GTU 57 | TIGSU 29 | *p*-Value |
|---|---|---|---|---|---|---|---|---|---|
| Analytical problem-solving skills | 2.44 | 321 | 0.720 | 2.66ab | 2.65cd | 2.32 | 1.98ac | 1.97bd | *** |
| Creative problem-solving skills | 2.52 | 321 | 0.700 | 2.69ab | 2.84def | 2.26d | 2.04ae | 2.07bf | *** |
| Ability to work in a lab | 2.48 | 321 | 0.750 | 2.69ab | 2.69cd | 2.29 | 2ac | 2.07bd | *** |
| Ability to search for relevant information on the internet | 2.42 | 321 | 0.670 | 2.58ab | 2.59cd | 2.23 | 2.18ac | 1.9bd | *** |
| Communication skills | 2.51 | 321 | 0.670 | 2.64ab | 2.74cd | 2.39 | 2.23ac | 1.97bd | *** |
| Team-working skills | 2.49 | 321 | 0.690 | 2.56abc | 2.8adef | 2.29d | 2.19be | 2.1cf | *** |
| Ability to adapt/act in new situations | 2.52 | 321 | 0.720 | 2.76abc | 2.76def | 2.23ad | 2.04be | 2.07cf | *** |
| Ability to innovate and create | 2.52 | 321 | 0.720 | 2.76abc | 2.77def | 2.29ad | 2.04be | 1.97cf | *** |
| Possessing basic knowledge | 2.49 | 321 | 0.690 | 2.68ab | 2.66cd | 2.29 | 2.18ac | 2.03bd | *** |
| Ability to compare and analyse different opinions | 2.46 | 321 | 0.740 | 2.7abc | 2.69def | 2.16ad | 2.02be | 1.93cf | *** |
| Ability to make judgements and justify decisions | 2.52 | 321 | 0.720 | 2.75abc | 2.77def | 2.29ad | 2.04be | 2cf | *** |

Notes: Respondents chose between 1 = not interesting, 2 = moderately interesting, 3 = very interesting. Kruskal–Wallis tests: *** marks $p < 0.0001$; letters indicate significant differences between universities (post hoc Games–Howell tests).

ATSU students were most interested in creative problem-solving skills and teamwork skills, more so than the other universities. AUG students, on the other hand, were most interested in the ability to innovate and create and the ability to adapt or act in new situations, showing significant differences from BSRSU, GTU, and TIGSU students. For TIGSU students, the ability to search for relevant information on the internet was the least interesting. These results suggest that students may find skills or competencies less interesting if they have already taken up training in these competencies in their past coursework. Meanwhile, those abilities they find most interesting may be unfamiliar to them or have not existed in their past coursework such that they find them important to acquire. Higher education institutions must be able to ensure the development of the skills that not only empower students but also equip them to deal with complex

issues and handling uncertainties as future professionals and principal actors in the food system [7,10,26].

Amongst the different teaching methods, Georgian students show the highest interest in "international courses", "seminars/interactive workshops", and "lectures with discussions", while being least interested in "e-learning courses" (Table 10). Significant differences were observed among the five Georgian universities. AUG and ATSU students were notably more interested in considering several teaching methods. AUG students preferred seminars and interactive workshops the most, while ATSU and BSRSU students preferred lectures with discussions most. On the other hand, GTU and TIGSU students preferred "international courses" the most. This observation may be explained by the fact that Georgia is a popular destination for international students. During the past several years, Georgia has increasingly gained more international students, which, in turn, has helped Georgian universities invest in infrastructure, technology, and the development of new educational programmes [27]. This presents a potential opportunity for Georgian universities to equip students with knowledge of SFS and increase their level of food citizenship to a wider extent, not only among Georgian students but among international students, too.

**Table 10.** Interests and preferences of Georgian students regarding different teaching methods.

| | | | | AUG | ATSU | BSRSU | GTU | TIGSU | |
|---|---|---|---|---|---|---|---|---|---|
| | Total Mean | N | SD | 114 | 87 | 31 | 57 | 29 | *p*-Value |
| Regular lectures | 2.19 | 321 | 0.700 | 2.15abc | 2.52ade | 2.29f | 1.79bdf | 2e | *** |
| Lectures with discussion | 2.50 | 321 | 0.680 | 2.62ab | 2.74cd | 2.48d | 2.14ac | 2.07bd | *** |
| Seminars/interactive workshops | 2.50 | 321 | 0.670 | 2.7ab | 2.64cd | 2.35 | 2.16ac | 2.1bd | *** |
| Group work | 2.41 | 321 | 0.730 | 2.47ab | 2.73acd | 2.35 | 1.95bc | 2.17d | *** |
| International courses (multi-cultural, international environment) | 2.54 | 321 | 0.610 | 2.69a | 2.69b | 2.32 | 2.21ab | 2.38 | *** |
| E-learning courses | 2.08 | 321 | 0.680 | 1.96a | 2.24a | 2.10 | 2.00 | 2.28 | |
| Cooperation with schools (e.g., giving lectures by students to school pupils) | 2.28 | 321 | 0.790 | 2.4a | 2.53bc | 2.35d | 1.77abd | 2c | *** |

Notes: Respondents chose between 1 = not interesting, 2 = moderately interesting, 3 = very interesting. Kruskal–Wallis tests: *** marks $p < 0.0001$; letters indicate significant differences between universities (post hoc Games–Howell tests)

Like Western European students, Georgian students found "e-learning courses" to be the least interesting, which may be attributed to students' misconception that e-learning courses often involve a highly passive learning method whilst they may be highly collaborative and thus highly appreciated by students [7,28]. Correlation analyses show that students who prefer lectures with discussions as a teaching method are also interested in seminars/interactive workshops (r = 0.704), group work (r = 0.633), multicultural/international environments (r = 0.627), cooperation with schools (r = 0.584), and regular lectures (r = 0.440). No negative correlations were observed between the teaching methods.

These results show that Georgian students prefer traditional teaching methods such as lectures while having simultaneous discussions and collaborative activities with other students from other schools or in international environments. These discussions enable students to attain "a critically informed understanding of the topic, self-awareness and capacity for self-critique, appreciation of diversity, and informed action" [20,29,30]. These exchanges serve as channels for students to combine theoretical concepts and practical examples while developing perspectives in interacting with international students. Moreover, this method of learning complements the skills sought by Georgian students, such as creative problem-solving skills, the ability to adapt/act in new situations, and the ability to innovate and create. Through eliciting creativity and collaboration among students, transdisciplinary and transactional learning may further develop these skill sets [7,20] and develop their collaborative skills at the same time. In order to ensure the implementation

of effective food systems programmes in Georgia, these learning methods should therefore serve as a basis for improvement of existing contemporary modules.

## 4. Study Limitations

Although the researchers intended to reach a representative part of all agriculture and food science students studying at Georgian universities, the two phases of the study were conducted during the COVID-19 pandemic, which may have limited the population sizes of the survey. This may be attributed to the lack of access to the internet, the unavailability of personal computers, or the lack of motivation to participate in any school-related activities among students during the confinement. Another limitation of the study is that the survey did not account for the nationality of the respondents, especially given that Georgian universities cater to a large population of international students. Moreover, the researchers were not able to access registrar data from the Georgian universities (to determine whether students lived with their families in the same city as their university). These factors may have influenced students' food preferences and purchasing behaviour. Finally, even though the selected universities all offer food and agriculture science degree programs, they vary largely in their foci and cover different specializations. This may have affected students' background knowledge, learning preferences, and behaviours. Ideally, the Higher Education Survey was to be shared and disseminated by professors/academic partners to their students on-site, yet this was not feasible since university instruction in Georgia during the data-gathering period was carried out by remote learning in accordance with pandemic restrictions. The food industry survey, on the other hand, was intended to be administered through in-depth qualitative expert interviews, yet due to travelling restrictions outside the EU and since the research program finished at the end of 2021, the researchers opted to administer the survey online through a structured questionnaire, therefore obtaining a limited population size.

## 5. Conclusions

In Georgia, as a developing country currently with a transitional economy, higher education and the food industry both play equally important roles in the development of Sustainable Food Systems. Notably, Georgian universities play a key role in the formation of specific competencies among the next generation of professionals in the country's transition to sustainability. The private sector, on the other hand, reinforces this transition through investment in smallholders and by providing an infrastructure for sustainable innovations and therefore reforming industry norms. Students' roles as future actors within the food system are not only established by their professional activities but also by their behaviours and attitudes as consumers. The results of providing a picture of sustainability from five agricultural universities in Georgia and of the Georgian food industry may provide pathways for the improvement of HE curricula with Sustainable Food Systems at their core.

H1—Georgian students' and food industry representatives' background knowledge of SFS and their topics. Most of the students have already received training or courses covering Sustainable Food Systems and its topics. These courses mainly cover the topics concerning "Food security" and "Food safety". The Georgian food industry representatives on the other hand are also generally knowledgeable about the subject but to a lesser extent. In terms of instruction and professional training, not all universities in Georgia which offer food and agriculture degree programs cover the identified topics. Likewise, not all food industry professionals have received training in Sustainable Food Systems. This suggests that both academic and professional training in Georgia need to develop pedagogical strategies to incorporate sustainability at the core of their instruction and business.

H2—Student's behaviours and food citizenship are positively influenced by their background knowledge of Sustainable Food Systems. A clear relationship between students' purchasing motives related to SFS was also established.

H3—The maintenance of healthy ecosystems was considered most important by students and professionals alike. Students who received more training in SFS considered

more the needs of future generations, that the food system should be humane and just and had more respect for animal welfare. This may, therefore, serve as a basis for HEIs to better form their students as future enablers of SFS while serving as a corporate social responsibility framework for the private sector.

H4—Students' preferences with respect to teaching methods and expectations in developing skills and competencies based on SFS. The results of this study demonstrate that the majority of Georgian students believe that Sustainable Food Systems topics would be useful for their future careers. The most-sought skills are problem-solving, the ability to adapt/act in new situations, the ability to innovate and create, and the ability to make judgements and justify decisions. Moreover, students' most preferred methods of teaching are international courses, seminars/interactive workshops, and lectures with discussions. Thus, these competencies and methods along with the topics of interest (organic agriculture and agroecology) should serve as a basis for the development of future curricula in Georgian higher education.

H5—The most interesting topics in SFS for students and professionals alike are organic agriculture and agroecology. Students who considered organic agriculture as an interesting topic considered environmental impacts, animal welfare, health, and local production in their purchasing decisions. Moreover, the interest of both students and professionals in organic agriculture and agroecology is reflective of the country's present focus and continuous efforts in developing standards to conform with EU regulations. In total, these points of interest may serve as guidelines for instruction in Georgian HEIs and as templates for sustainability strategies or initiatives for the Georgian food industry.

## 6. Implications for Future Research

The findings of the study represent groundbreaking research, providing an overview of Sustainable Food Systems in Georgian higher education and the food industry. The results show a comparative overview of SFS education among five agricultural universities and thereby reveal pathways in remodelling education programmes towards SFS. These insights may also serve as an empirical basis for interventional studies that aim to improve food citizenship among students and food professionals alike. Since the study only focused on food purchasing and preparation performed by students, future work may focus on examining sustainable food consumption behaviours in more detail while considering whether students live with their families or not. Finally, future studies focusing on Georgian higher education and the food industry may consider the use of a mixed-method approach (i.e., explanatory design) to further explain the mechanisms behind the resulting variables and to address new questions arising from the quantitative phase.

**Author Contributions:** Conceptualization, A.E. and A.P.; methodology, T.G.; software, A.E.; validation, A.P., A.E. and T.U.; formal analysis, all authors.; investigation and questionnaire translation, T.G., A.E. and A.P.; data curation, A.E.; writing—original draft preparation, A.E.; writing—review and editing, all authors; visualization, A.E.; supervision, A.P. and T.U.; project administration, all authors; funding acquisition, A.P. All authors have read and agreed to the published version of the manuscript.

**Funding:** This research received no external funding.

**Informed Consent Statement:** Informed consent was obtained from all subjects involved in the study.

**Data Availability Statement:** The data presented in this study and survey questionnaires are available at: https://doi.org/10.6084/m9.figshare.19285475.v2 (accessed on 3 March 2022).

**Conflicts of Interest:** The authors declare no conflict of interest.

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
