# Peer review of "Knowledge and Expectations Regarding Sustainable Food Systems among Students from Georgian Agricultural Universities and Georgian Food Industry Representatives"

_sustainability, doi:10.3390/su14095128_

Round 1

Reviewer 1 Report

This manuscript presents intriguing research on sustainable food systems (SFS). The authors have collected detailed data from Georgian students and food industry representatives and analyzed it to explore their knowledge and expectations on SFS. The article fits the journal's scope, the writing is clear, the results are valuable, and the authors have also elaborated on the limitations of the research. However, some issues still need to be addressed. Please see my comments below.

  • Please remove the headings from the background. Follow the journal guidelines for writing the abstract. Revise the lines 13-15. Also, please avoid using too many abbreviations in the abstract.
  • Line 23: Please correct this line.
  • Keywords should be precise, relevant to the article's content, and different from those provided in the title. Please revise.
  • Lines 57-65 should be contextualized in the overall narrative being built in the introduction. Please avoid providing recommendations at the outset.
  • Please review some recent literature on the issue. See for example:S
    • Saqib, Zulkaif Ahmed, et al. "Education for sustainable development in Pakistani higher education institutions: An exploratory study of students' and teachers' perceptions." International Journal of Sustainability in Higher Education (2020).
  • The hypothesis should be generally based on a solid theoretical foundation which is lacking in the case of your hypotheses. Please improve this part. Also, it is suggested that the authors provide a relevant literature review (Georgian context) before proposing the hypotheses.
  • Why were the students chosen only from agricultural universities? Also, please indicate this in the title of your paper.
  • Please provide a Table of descriptive statistics of respondents in section 3.1.1.
  • Please be consistent in the formatting of the article. The caption of Table 3 should be above the Table.
  • Please provide Figure 1 in line with the text. Follow the journal's format.
  • Instead of providing only Pearson correlations, it is suggested that the authors include a detailed econometric analysis to identify the factors influencing the outcome variables. Furthermore, it should be explored why students' responses vary across different universities.
  • Figure 3 and 4: It is suggested to convert these statistics into percentages for a clear understanding.
  • The findings related to food industry representatives should be explained in more detail, and also please enhance the relevant methodology.

Author Response

Kindly see the attachment.

Reviewer 2 Report

When reviewing scientific papers for publication, I usually start with a general overview in terms of a structure, abstract, literature review, methodology, findings of the research, discussion, conclusions, as well as limitations of the study and future directions of the research. I also pay attention to the language level, especially if the paper is written in English, and English is not the native language. 

The reviewed paper entitled “Knowledge and Expectations on Sustainable Food Systems among Georgian Students and Food Industry Representatives” is generally structured in a proper way. There is, however no section "future directions of the research”. This section should be added too, given this is a research paper.

The literature review is average but is founded in the existing literature of the topic. Generally I claim that Author (s) provide theoretical foundations for the analysis using appropriate references. I would, however, recommend to add much more references  devoted to the latest literature associated with the topic in question (including Web of Science and Scopus papers).

The research methodology should be presented more precisely. We have no precise information on how the research sample was selected. Was the sample representative? The presented material shows that rather not - it means that statistical analyzes should not be carried out on the basis of the collected data.  What was the research procedure? There is no questionnaire in the attachment.

 A weak point of this paper is "Discussion" section. I suggest two separate sections "Results" and "Discussion". "Discussion" should discuss the results achieved; In addition, there should be references to the results of other scholars. Unfortunately we have not too much in this part, and the second aspect is missing also. Discussion is an interpretation of the results – implications, significance of results. Provide the response to the research question(s). Interpret results taking into account alternative explanations - where applicable. What are the practical implications (and theoretical –where applicable) suggested by the results of your research. New questions which emerge from your research. Be careful not to “go beyond” your data and results, in particular if the focus of your study is narrow. You can “suggest”, or even “speculate” in the discussion, but it must be clearly evident what is derived from a result and what is your suggestion, comment or speculation,  ...

I also recommend a final proofreading of the paper to be done by the native speaker. 

Author Response

Kindly see the attachment.

Round 2

Reviewer 1 Report

The authors have made significant changes to the manuscript that have improved its quality. The manuscript has now reached the desired standard of scientific publication, and I am pleased to recommend its acceptance. Congratulations to the Authors. 
Some Minor Changes:

- The word "non-state actors" should be removed from Line 68 and all other places where it appears. It should be replaced with another suitable term. 

- In Section 6 (Implications), remove the word "groundbreaking research" and tone it down a bit. 

Reviewer 2 Report

Thank you for the corrections made. The article is currently much better and suitable for publication.